# Hallmarks of Basidiomycete Soft- and White-Rot in Wood-Decay -Omics Data of Two *Armillaria* Species

**DOI:** 10.3390/microorganisms9010149

**Published:** 2021-01-11

**Authors:** Neha Sahu, Zsolt Merényi, Balázs Bálint, Brigitta Kiss, György Sipos, Rebecca A. Owens, László G. Nagy

**Affiliations:** 1Biological Research Center, Synthetic and Systems Biology Unit, 6726 Szeged, Hungary; nehasahu486@gmail.com (N.S.); zmerenyi@gmail.com (Z.M.); h836472@gmail.com (B.B.); kiss.brigitta84@gmail.com (B.K.); 2Doctoral School of Biology, Faculty of Science and Informatics, University of Szeged, 6726 Szeged, Hungary; 3Research Center for Forestry and Wood Industry, Functional Genomics and Bioinformatics Group, University of Sopron, 9400 Sopron, Hungary; gyoergy.sipos@wsl.ch; 4Swiss Federal Research Institute WSL, Zürcherstrasse 111, CH-8903 Birmensdorf, Switzerland; 5Department of Biology, Maynooth University, W23 F2H6 Kildare, Ireland; Rebecca.Owens@mu.ie; 6Department of Plant Anatomy, Institute of Biology, Eötvös Loránd University, 1117 Budapest, Hungary

**Keywords:** *Armillaria* wood-decay, basidiomycete soft-rot, plant biomass degradation, plant pathogens, proteomics, RNA-Seq, wood-decay types

## Abstract

Wood-decaying Basidiomycetes are among the most efficient degraders of plant cell walls, making them key players in forest ecosystems, global carbon cycle, and in bio-based industries. Recent insights from -omics data revealed a high functional diversity of wood-decay strategies, especially among the traditional white-rot and brown-rot dichotomy. We examined the mechanistic bases of wood-decay in the conifer-specialists *Armillaria ostoyae* and *Armillaria cepistipes* using transcriptomic and proteomic approaches. *Armillaria* spp. (Fungi, Basidiomycota) include devastating pathogens of temperate forests and saprotrophs that decay wood. They have been discussed as white-rot species, though their response to wood deviates from typical white-rotters. While we observed an upregulation of a diverse suite of plant cell wall degrading enzymes, unlike white-rotters, they possess and express an atypical wood-decay repertoire in which pectinases and expansins are enriched, whereas lignin-decaying enzymes (LDEs) are generally downregulated. This combination of wood decay genes resembles the soft-rot of Ascomycota and appears widespread among Basidiomycota that produce a superficial white rot-like decay. These observations are consistent with ancestral soft-rot decay machinery conserved across asco- and Basidiomycota, a gain of efficient lignin-degrading ability in white-rot fungi and repeated, complete, or partial losses of LDE encoding gene repertoires in brown- and secondarily soft-rot fungi.

## 1. Introduction

Fungi play an immense role in terrestrial ecosystems as decomposers of organic matter, plant pathogens, and mutualistic symbionts. Being the primary decomposers of woody organic matter in forest ecosystems, they contribute substantially towards global carbon cycling by releasing sequestered organic carbon back to the soil. They do so by producing a wide arsenal of plant cell wall degrading enzymes (PCWDEs), which can decay all components of the plant biomass including the recalcitrant lignin, and thus can be used for sustainable utilization of the biomass waste [1,2,3,4,5]. Recent fungal -omics studies involving basidiomycetes revealed substantial differences in the plant cell wall degrading enzyme (PCWDE) repertoires of fungi. Genomic patterns align well with fungal lifestyles, including the traditional, broad classification of decay types into either white-rot (complete degradation of plant cell wall components) or brown-rot (lignin remains unmodified or slightly modified) [6,7,8]. However, how finer-scale differences in wood decay gene repertoires translate to functional heterogeneity is not known. Exploring the functional and taxonomic diversity of wood decay fungi could provide opportunities for better understanding of forest ecosystems and for identifying fungal enzymes of biotechnological potential.

*Armillaria* spp. (Basidiomycota, Agaricales, Fungi) are among the most devastating fungal pathogens in woody ecosystems, including temperate forests, tree plantations, vineyards, and gardens [9,10] and are known to cause tremendous losses to the economy, health, and long-term productivity of forests [11,12,13,14]. The genus *Armillaria* comprises about 70 known species [15] including both pathogens and saprotrophs, making them suitable for studying mechanisms of pathogenicity and wood-decay systems in fungi [11,14,15,16,17].

*Armillaria* spp. have immense ability to degrade plant biomass, and thus are ecologically significant for releasing sequestered organic C into the carbon cycle. Previous comparative genomic studies on *Armillaria* highlighted plant cell wall degrading enzyme repertoires reminiscent of white-rot fungi [15,17,18,19], with a characteristic enrichment of pectinolytic genes [17,19]. Accordingly, recent studies treated *Armillaria* spp. as white-rot based on the presence of LDE encoding genes in their genomes [15,17,19,20,21]. However, previous studies have also shown that *Armillaria* species primarily decay the cellulose, hemicellulose, and pectin components of the plant cell wall, and leave lignin unattacked during early stages of decay [22,23]. Chemical and microscopic analyses of wood decay by *Armillaria* produced contradictory results. *A. mellea* was classified as Group II white-rot fungi which attack celluloses and pentosans at early stages and lignin remains unaffected [23]. Others reported a type-I soft-rot decay where the fungal hyphae grow through the secondary cell wall layer, producing characteristic cavities in the tracheids, axial, and xylem ray parenchyma cells [22]. Soft-rot fungi by definition are now restricted to Ascomycota [24,25]; however, there are many Agaricomycetes that produce symptoms resembling soft rot or that do not fit the traditional white rot/brown rot dichotomy [7,8,20]. Soft rot was suggested in *Cylindrobasidium* spp., a close relative of *Armillaria,* that also has a lower number of LDEs than typical white-rot decayers [17,20]. 

White rot fungi remove lignin from wood using high-redox potential oxidoreductases (e.g., class-II peroxidases [6,26]) and degrade complex polysaccharide polymers using diverse glycosyl hydrolase (GH), auxiliary activity (AA), carbohydrate esterase (CE). and polysaccharide lyase (PL) cocktails [3,27]. Fungi from multiple white-rot and brown-rot clades have been used to study the wood-decay associated gene expression [28,29,30,31,32]. These studies highlighted substrate and species-specific responses, sequential activation of degradative enzymes, along with lifestyle-driven differences among species highlighting the ambiguity of extant decay-type classification. Previous studies on *Armillaria* have revealed their unusual gene repertoire [15,17], making them suitable candidates to understand where *Armillaria* fits in the ever-growing array of decay types.

We here employed a comparative multi -omics approach for understanding wood-decay by *Armillaria*, a group of wood-colonizing and pathogenic fungi that, in previous comparative genomic studies, showed great potential for improving our understanding of the functional diversity of wood-decay. We allowed a pathogenic (*A. ostoyae*) and a saprotrophic (*A. cepistipes*) species to colonize Norway spruce (*Picea abies*) roots, and performed RNA-Seq and proteomics on invasive mycelium and rhizomorphs and their non-invasive counterparts. We opted for a comparative, instead of a time-series analysis, to enable evolutionary comparisons between two species and emphasize species diversity in *Armillaria*. We found that the saprotroph *A. cepistipes* showed a stronger response to wood than *A. ostoyae*, suggesting adaptations to faster wood colonization in saprotrophs than in pathogens. The two species of *Armillaria* displayed a wood-decay profile that, unlike in white-rot fungi, mostly bypasses the activation of LDE encoding genes and is characterized by the upregulation of pectinolytic, cellulolytic, iron acquisition-related and expansin genes. These expression patterns resemble soft-rot, which, however, is selective and based on the transcriptional regulation of genes, given that *Armillaria* genomes encode LDE genes. Altogether, the decay profile of *Armillaria* highlights a widespread wood-decay strategy in the Agaricomycetes and contributes to unveiling the functional diversity of forest fungi.

## 2. Material and Methods

### 2.1. Species Choice, Wood Colonization Assay, and RNA-Extraction

*Armillaria ostoyae* is a pathogen while *A. cepistipes* is a saprotroph with a mortality ratio not exceeding 5% [16,33]. Both of them preferentially colonize conifers [16]. Cultures of *Armillaria ostoyae* C18 and *Armillaria cepistipes* B2 [17] were inoculated on Malt extract agar (MEA) and incubated at 25 °C in dark for a week. Next, 4–5 cm long, autoclave sterilized spruce (*Picea abies*) roots were introduced to the week-old cultures and were kept at 25 °C in the dark until the roots were completely colonized by mycelial sheath and rhizomorphs were growing out of the it. Sampling was performed at the time when rhizomorphs were seen emerging out of the roots (2–3 weeks) (Figure 1B). After colonization, roots were dissected to collect mycelium and rhizomorphs emerging out of the colonized wood (Figure 1B). Cultures of *Armillaria* without the addition of spruce roots were used as controls. The tissues were snap frozen in liquid nitrogen and kept at −80 °C till further use. Total RNA was extracted in three biological replicates by homogenizing the tissues using mortar and pestle, followed by processing samples using the Quick-RNA Miniprep kit (Zymo Research, Irvine, CA, USA), following the manufacturer’s protocol.

### 2.2. RNA-Seq Library Generation and Sequencing

RNA-Seq analyses were carried out by using Ribo-Zero rRNA removal kit (Yeast) to deplete rRNA from total RNA. Subsequently, samples were processed with Illumina TruSeq V2 library preparation protocol. Libraries were sequenced on an Illumina NextSeq 500 machine yielding 2 × 150 nt reads.

### 2.3. Protein Extraction

For protein extraction, tissues were snap-frozen in liquid N_2_ and bead beaten (Retsch Mixer Mill MM400) periodically (30 Hz, 2 min), with snap freezing between cycles. Lysis buffer (6 M Guanidine-HCl, 0.1 M Tris-HCl, 50 mM DTT pH 8.6) was added to crushed fungal tissue and bead-beating was repeated. Samples were further disrupted using sonication (MS72 probe, 3 × 10 s), with cooling on ice between sonications. Samples were clarified by centrifugation and supernatants passed through 3 kDa cut-off filters (Millipore) to concentrate and perform buffer exchange into PBS. Protein samples were precipitated with TCA (final 15% *w/v*) and pellets were washed with ice-cold acetone. Protein pellets were resuspended in UT buffer (6 M Urea, 2 M Thiourea, 0.1 M Tris-HCl pH 8) and concentrations normalized following Bradford protein assay. Samples were digested according to Moloney et al. [34] and ZipTips (Millipore, Billerica, MA, USA) were used for sample clean-up. Peptide samples were analyzed using the high mass accuracy Q-Exactive mass spectrometer (ThermoFisher Scientific) coupled to a Dionex Ultimate 3000 nanoLC with an EasySpray PepMap C18 column (50 cm × 75 µm). Peptide mixtures were separated as described in Collins et al. [35] and resultant data were analyzed using MaxQuant (v 1.5.3.30) [36] with the label-free quantitation (LFQ) algorithms and searching against the protein database (filtered models) in JGI MycoCosm [17].

### 2.4. Bioinformatic Analyses of RNA-Seq Data

Paired-end Illumina (HiSeq, NextSeq, Seqomics Kft., Mórahalom, Hungary) reads were quality trimmed using the CLC Genomics Workbench tool version 11.0 (CLC Bio/Qiagen) removing ambiguous nucleotides as well as low quality read ends. The quality cutoff (error probability) was set to 0.05, corresponding to a Phred score of 13. Trimmed reads of >40 bases were mapped using the RNA-Seq Analysis 2.16 package in CLC requiring >80% sequence identity over >80% of the read length; strand specificity was omitted. Reads with less than 30 equally scoring mapping positions were mapped to all possible locations while reads with >30 potential mapping positions were considered as uninformative repeat reads and were excluded. “Total counts” RNA-Seq count data was imported from CLC into R version 3.0.2. We kept only genes detected by at least five mapped reads in at least 25% of the samples. Subsequently, “calcNormFactors” from “edgeR” version 3.4.2 [37] was used to perform data scaling based on the“trimmed mean of M-values” (TMM) method [38]. Log transformation was carried out by the “voom” function of the “limma” package version 3.18.13 [39]. Linear modeling, empirical Bayes moderation as well as the calculation of differentially expressed genes were carried out using “limma”. Genes showing an at least two-fold gene expression change with an FDR value <0.05 were considered as significant. Multidimensional scaling (“plotMDS” function in edgeR) was applied to visually summarize gene expression profiles. In addition, unsupervised cluster analysis with Euclidean distance calculation and complete-linkage clustering was carried out on the normalized data using “heatmap.2” function from R package “gplots”.

### 2.5. Analyses of Proteomic Data

Proteomic results were analysed using Perseus (v 1.5.4.0) [40]. Qualitative and quantitative analyses were performed to determine the relative changes in protein abundance. Quantitative analysis was performed using a student’s t-test with a *p*-value cut-off of 0.05 and log_2_(fold change; FC) ≥ 1. Qualitative analysis revealed proteins that were detected in 2/3 replicates for a sample type and undetectable in all replicates of the comparator group. A theoretical minimum fold change was determined for qualitative results based on a calculated minimum detectable protein intensity (mean + 2 standard deviations of lowest detectable protein intensity for each replicate in the experiment) [34]. Based on this theoretical minimum fold change, some qualitative results were excluded due to intensity values approaching the minimum detectable levels. Qualitative and quantitative results were combined, and a total number of differentially abundant proteins (DAPs) are summarized in Appendix A.

### 2.6. Clustering and Functional Annotation

Orthologous proteins in *A. ostoyae* and *A. cepistipes* were identified by OrthoFinder v2.3.1 [41] (default parameters). Single-copy orthologs for the two species (Appendix A) were further analyzed. Functional annotation was done based on Interpro domains using InterProScan v5.24-63.0 [42].

Enriched GO terms were identified by topGO [43], using the weight01 algorithm and Fisher testing. Terms with *p*-values < 0.05 were considered significant.

The CAZyme copy numbers in *A. ostoyae* and *A. cepistipes* were collected from the JGI Mycocosm annotations, which are based on the CAZy annotation pipeline [44]. We separated the CAZy families based on their substrate-specificity (Appendix A).

Transporters were identified by using DeepLoc [45] to select plasma membrane-localized proteins from the proteomes of both species. Plasma membrane-localized proteins with >1 transmembrane domains were used to obtain a list of non-redundant InterPro domains, which were manually curated for functional roles in transport.

## 3. Results

### 3.1. Morphological Observations and Type of Samples

Sterilized Norway spruce roots were introduced to one-week-old cultures of *A. ostoyae* and *A. cepistipes*. We observed abundant growth of the mycelium in and under the bark layer (Figure 1B). Although previous studies suggested that colonization happens via direct rhizomorph contact and penetration [11,13,15,17,46,47,48,49], we did not find evidence for the mechanical entry of rhizomorphs into the bark. Instead, upon contact with the root, rhizomorphs switch to hyphal growth (Appendix A). Rhizomorphs were not observed below the bark in *A. ostoyae* even after eight weeks of incubation (roots were completely colonized by the fungus by four weeks, growth rate of *Armillaria mellea* mycelium is around 0.75 mm/day [50]), whereas *A. cepistipes* grew rhizomorph-like structures below the bark layer (Appendix A). Both species exited the root section as rhizomorphs. These observations suggest that mycelium is the primary colonizing structure of wood, which is consistent with the higher surface/volume ratio of hyphae being better suited for nutrient acquisition as compared to rhizomorphs. Rhizomorphs probably emerge much later, possibly to transfer nutrients, as seen commonly under the bark of decayed logs [11].

### 3.2. Overview of New-Omics Data

We analyzed four tissue types from both species in three biological replicates using transcriptomics and proteomics (Figure 1B). The mycelium and rhizomorphs collected from colonized roots are hereafter referred to as invasive mycelium (M) and invasive rhizomorphs (R), while those grown in the absence of roots are referred to as non-invasive mycelium (NIM) and non-invasive rhizomorphs (NIR), respectively. This yielded 12 samples for both *A. ostoyae* and *A. cepistipes* (with an additional sample type in the latter, see Appendix A). We prepared ribosomal RNA-depleted libraries and sequenced them to a depth of 46.7–93.4 million paired-end reads on the Illumina NextSeq 500 platform. On average, 69% and 38% of the reads mapped to the transcripts in *A. ostoyae* and *A. cepistipes*, respectively (Appendix A). Regarding the low mapping percentages in *A. cepistipes,* we find that they were either caused by genomic DNA contamination or a poor annotation of the reference species (~50% of unmapped reads mapped to intergenic regions, and not transcripts). Although such factors can dampen the signal of differential expression, we find that in our case this did not compromise our analysis of differential gene expression (see the clear separation of samples in the MDS, Figure 1C,E).

Label-free comparative proteomics provided relative abundance of data across the different sample types. A total of 37,879 and 36,087 peptides were identified from *A. ostoyae* and *A. cepistipes,* respectively, which were subsequently rolled up into protein groups, with median protein sequence coverage ranging from 30.1 to 34%.

Multidimensional scaling (MDS) plots show a strong clustering of the biological replicates in both transcriptomic and proteomic data in both species. The MDS plots portray a clear separation of the invasive and non-invasive mycelium samples, whereas invasive rhizomorphs and non-invasive rhizomorphs showed higher similarity to each other (Figure 1C for *A. ostoyae*, Appendix A for *A. cepistipes*), suggesting that the larger difference exists between invasive and non-invasive mycelium. For *A. ostoyae* and *A. cepistipes,* proteomic analyses detected 3177 and 3232 proteins, respectively (Figure 1D). About 99.8% of the detected proteins were also present in the RNA-Seq data (Figure 1D). At the same time, there was a limited correlation between the fold-change values acquired for transcripts and proteins (Appendix A), which is not unusual among proteomic and transcriptomic datasets.

In *A. ostoyae* we found 987 and 35 significantly upregulated genes (log_2_FC > 1, *p*-value < 0.05) in invasive vs. non-invasive mycelium (M*vs*NIM) and in invasive vs. non-invasive rhizomorphs (R*vs*NIR) (5.74% and 0.2% of transcriptome), respectively. Considerably more, 2108 and 327 upregulated genes were found to be differentially expressed in M*vs*NIM and R*vs*NIR in *A. cepistipes*, respectively (13.29% and 2.06% of transcriptome; Appendix A). The number of differentially abundant proteins (DAPs) in the proteomics analysis was lower: we detected 279 and 108 proteins with increased abundance in MvsNIM and RvsNIR in *A. ostoyae* (8.78 and 3.40 % of detected proteome), and 439 and 282 proteins with increased abundance in MvsNIM and RvsNIR, respectively, in *A. cepistipes* (13.58 and 8.73 % of detected proteome; Appendix A).

We found considerably more DEGs/DAPs in the mycelium than in rhizomorphs (Figure 1E) in both species, which indicates that the mycelium is more actively involved in the colonization of woody tissues than are rhizomorphs. A more surprising observation is that the saprotrophic *A. cepistipes* shows a higher number of DEGs/DAPs than the pathogenic *A. ostoyae* (Figure 1E). To confirm that this is not a result of higher baseline expression of some genes in *A. ostoyae*, we compared the distribution of raw expression values of co-orthologs in the M and NIM samples in both species (Appendix A). This showed that baseline expression of genes in non-invasive mycelia of *A. ostoyae* was not higher than that in *A. cepistipes*, indicating that the higher number of DEGs in *A. cepistipes* is indeed the result of the stronger reaction of this species to wood. We speculate that this is because saprotrophs, to gain a competitive advantage over other microbes, have to colonize/degrade wood faster than necrotrophic pathogens, such as *A. ostoyae*, which can both feed on living parts of the tree and, upon killing the host, can be the first colonizers of the wood [17,51].

### 3.3. Gene Ontology (GO) Analyses

In *A. ostoyae*, we found 25 and 11 GO terms significantly enriched among the genes upregulated in M*vs*NIM and R*vs*NIR, respectively (Figure 2A). For *A. cepistipes* we found 57 and 29 terms enriched in M*vs*NIM and R*vs*NIR, respectively (Appendix A). In both species, genes upregulated in invasive mycelia were enriched (*p* < 0.05, Fisher’s exact test) for terms related to oxidation-reduction, lipid metabolism, transmembrane transport, iron ion, and heme-binding processes along with the terms "fungal type cell wall" and "structural constituent of cell wall". We find that the enrichment of iron-binding related terms was driven by the upregulation of members of the high-affinity iron permease complex (2 out of 3 genes and, 1 out of 2 genes upregulated in *A. ostoyae* and *A. cepistipes,* respectively). Consistent with the lower number of DEGs in invasive vs non-invasive rhizomorphs, we observed fewer enriched GO terms (see Appendix A).

We observed similar patterns in the proteomic data: there were more enriched GO terms in *A. cepistipes* than *A. ostoyae*. In *A. ostoyae*, we found 18 and 12 significantly enriched terms in M*vs*NIM and R*vs*NIR, respectively (Figure 2B), whereas, in *A. cepistipes*, there were 35 significantly enriched terms in M*vs*NIM and 19 in R*vs*NIR (Appendix A). Terms enriched in M*vs*NIM included pectinesterase activity, polygalacturonase activity, cellulose-binding, carbohydrate-binding, hydrolase activity, together with cell wall modification, cell wall-related terms, and sugar metabolic process.

### 3.4. Global Transcriptome and Proteome Similarity

We measured global similarity among transcriptomes and proteomes within and across species based on Pearson correlation. In general, we observed a better correlation among transcriptomes than among proteomes (Figure 3).

Within species, we observed limited differences among transcriptomes, with the highest global transcriptome similarity values observed between invasive tissue types (Figure 3A). For example, in *A. ostoyae*, the two most similar sample types were invasive mycelium and invasive rhizomorphs (mean Pearson: 0.92), slightly higher than other combinations of samples (0.83–0.88). A similar, but a stronger pattern is observable in *A. cepistipes* (Figure 3A). This pattern suggests that contact with wood elicits similar expression changes irrespective of the tissue type. In support of this, we could identify 4 and 127 genes upregulated in the invasive mycelium and invasive rhizomorphs of *A. ostoyae* and *A. cepistipes*, respectively. Many of these genes were annotated as hydrophobins, cytochrome P450s, galactose-binding domain-like proteins, and a number of CAZymes (Appendix A).

The among-species similarity between sampled tissues was assessed based on 11,630 co-orthologous genes in *A. ostoyae* and *A. cepistipes*, identified by OrthoFinder [41], out of which transcriptomic and proteomic data cover 10,675 and 2404, co-orthologs, respectively. A surprisingly high correlation was found between the invasive mycelia of *A. ostoyae* and *A. cepistipes* (Appendix A*)*, whereas the correlation was comparatively lower in all other combinations. This observation was similar for both transcriptomic and proteomic data and was even more pronounced when we considered only genes/proteins that were DEG or DAP in at least one of the species (Figure 3B). We interpret the correlated gene expression in *A. ostoyae* and *A. cepistipes* as an indication of a shared response of invasive mycelia to the presence of spruce roots.

### 3.5. Shared Transcriptomic Response of Mycelia to Wood

To understand what comprises the observed similarity in wood decay, we focused on co-orthologous gene/protein pairs up- or downregulated in invasive mycelia of both species. We found 779 co-orthologs having similar differential expression in the invasive mycelium. Of these 372 and 407 were significantly up- and downregulated, respectively (Figure 4B, Appendix A). For the 372 upregulated co-orthologs, we observed overall higher fold changes and expression levels in *A. cepistipes* than in *A. ostoyae* (Figure 4B), again, underscoring a stronger response of *A. cepistipes* to wood.

Among the most upregulated co-orthologs in the transcriptomic analyses, we found oxoglutarate/iron-dependent dioxygenases, proteins of the galactose-binding-like domain superfamily (including CBM67, see below), ricin-B lectins, hydrophobins, intradiol-ring cleavage dioxygenases, GMC oxidoreductases, cytochrome p450-s, as well as 10 conserved transcription factors and several unannotated genes (Figure 4B, Appendix A). The most highly induced genes in both species were oxoglutarate/iron-dependent dioxygenases. These were reported to be responsible for the oxidation of organic substrates, mycotoxin production, and secondary metabolite biosynthesis [52,53,54] and were also found to be upregulated in both white-rot and brown-rot wood decay studies [55,56,57]. We found 46 oxoglutarate/iron-dependent dioxygenase genes in both species, of which five and nine were upregulated in the invasive mycelium of *A. ostoyae* and *A. cepistipes,* respectively (but not in invasive rhizomorphs). In proteomics, we found one and four genes in the invasive mycelium and one and two genes in the invasive rhizomorphs with increased abundance in *A. ostoyae* and *A. cepistipes,* respectively. The 2-oxoglutarate dioxygenase superfamily is widespread across microorganisms, fungi, plants, and mammals as well [54,58,59,60]; however, their versatile nature makes them difficult to interpret in terms of exact biological relevance in wood decay mechanisms.

In the proteomics data, we found 89 co-orthologs with increased and 45 with decreased abundance in both species (Figure 4C; Appendix A), of which, the ones with the highest abundance in both species included GH31, GH3, GH88, and GH92 CAZyme families as well as aspartic peptidases, fungal lipases, and Kre9/Knh1 fungal cell wall-related proteins. Some of these proteins were only detectable in the invasive mycelium and not in non-invasive mycelia, including several CAZymes such as pectin lyases, GH28, carbohydrate-binding modules (CBMs), PL8, GH3, GH35, galactosidases, carboxylesterases, and several other gene families like GMC oxidoreductases, various transporters, and cytochrome P450s.

### 3.6. Characteristic PCWDE Expression in Invasive Mycelia

A diverse array of plant cell wall degrading enzymes (PCWDEs) were found to be differentially expressed in the invasive tissues of both species. Overall the number of upregulated PCWDEs in the invasive mycelium was much higher than the invasive rhizomorphs when compared to their non-invasive counterparts (Figure 5). The saprotrophic *A. cepistipes* had a higher number of DEG/DAP PCWDEs than the pathogenic *A. ostoyae* (Appendix A; Figure 5). Among differentially expressed PCWDEs in mycelium vs non-invasive mycelium, upregulated pectinases were most numerous accounting for 17% and 37% of all pectinases in *A. ostoyae* and *A. cepistipes,* respectively. These were followed by cellulases (9%, 27%), hemicellulases (11%, 30%), and expansins (12%, 10%) (Figure 5; see Appendix A for the complete list of differentially expressed PCWDEs). We found most of the LDE related genes to be downregulated in both species, with none upregulated in *A. ostoyae* and few upregulated in *A. cepistipes* (Figure 5*)*. In addition to the general downregulation of LDE encoding genes, four copies of dye decolorizing peroxidases (DyPs) were present in each species, but none of these were detected in the wood-decay transcriptomics and proteomics in *A. ostoyae* and one only downregulated DyP was observed in invasive mycelium of *A. cepistipes*. We found one and eight upregulated LPMOs/AA9 in invasive mycelia of *A. ostoyae* and *A. cepistipes,* respectively, which might act together with the cellobiohydrolases to enhance cellulose degradation [61,62,63,64,65,66].

Our analyses revealed significant expression of both pectinolytic PCWDEs and expansins (Figure 5; Appendix A), indicating hallmarks of early-stage wood decay. In previous time-series studies, abundant pectinase expression was found during the early stages of wood-decay, suggesting a requirement of early-stage pectinolytic ‘pretreatment’ [67,68] for making the plant cell wall structure accessible, followed by a wave of non-pectinolytic GHs expression. In previous studies [67], it was also observed that early stages of wood decay were marked by increased expression of expansins and GH28 pectinases, suggesting both enzymatic and mechanical loosening of the plant cell walls for easier access of the cellulose and hemicellulose components. Our observations of pectinolytic PCWDE and expansin expression are in line with this.

In contrast, we detected few lignin-degrading AAs to be DEG/DAP in the two species, the majority of which were downregulated, suggesting that lignin was not appreciably attacked by *Armillaria* in our experiments. The chemical composition of spruce roots was reported to have higher lignin content (25–28%) followed by cellulose, hemicellulose, and extractives [69], yet our study revealed a general downregulation of LDEs. Of the 39 genes that encode LDEs (Appendix A) in each of the species, we found no upregulated and about 15% downregulated LDEs in *A. ostoyae*. In *A. cepistipes*, we found 15% upregulated and 30% downregulated LDEs. In the *A. cepistipes,* the proteomic data revealed somewhat more substantial lignin degradation: we found 4 (out of 11 detected) AA1_1 laccases, 3 AA2 peroxidases (out of 4), and 7 (aryl) alcohol dehydrogenases (AA3_2, of 17) increased in invasive relative to non-invasive mycelium. The modest induction and general downregulation (in the transcriptomic data) of LDE encoding genes are remarkable for white-rot fungi, especially in the light of previous studies reporting an early activation of LDEs by white-rot fungi [68,70] and the enrichment of class II peroxidases and other AAs (e.g., laccases) near the hyphal front. The lack of a lignin-modifying burst is, on the other hand, consistent with the underrepresentation of LDE related gene families in *Armillaria* compared to other white-rot Agaricales [17]. We note that heat-based lignin breakdown or loosening of the lignocellulose matrix during autoclaving [71,72,73] may provide a complimentary, although a less likely explanation of our data.

### 3.7. Evidence for Pectinolysis from Galactose Binding Domain Proteins

We found a high number of galactose-binding-like domain superfamily protein (GBDPs) genes upregulated the two species, especially in invasive mycelia. This protein superfamily includes, among others, the rhamnose-binding module family CBM67. Out of 84 and 89 genes containing GBDPs in *A. ostoyae* and *A. cepistipes,* we found 13 and 29 upregulated in invasive vs. non-invasive mycelia, respectively. Four of these genes were also among the 20 most highly induced co-orthologous genes in invasive mycelia (see Figure 4). In the proteomics data, *A. ostoyae* had 7 GBDPs (2 CBM67s) and *A. cepistipes* had 12 GBDPs (2 CBM67s) with increased abundance in the mycelium vs. non-invasive mycelium. A significant portion (35–50%) of upregulated genes were annotated as CBM67s in the CAZy database [44]. CBM67 are l-rhamnose binding modules, which are reported to be involved in pectin degradation [17,74]. Apart from CBM67, there were a number of other pectinolytic enzymes (e.g., GH78, PL4) associated with these GBDPs, which were upregulated in the two species (Appendix A). The abundance of l-rhamnose binding modules on their own, or in combination with pectinolytic enzyme encoding genes, as well as the dominance of pectinolytic PCWDEs among upregulated CAZymes could suggest a decay strategy focused on pectin removal for accessing the cellulose and hemicellulose units of the plant cell wall.

### 3.8. Iron Acquisition Genes Upregulated in Armillaria spp.

We observed a number of iron acquisition genes to be upregulated in mycelia vs. non-invasive mycelia of both species. Uptake of extracellular ferrous iron (Fe^2+^) occurs in fungi via a two-part transporter, which consists of an iron permease (Ftr1/FtrA) and an associated multicopper oxidase (Fet3) [75]. In *A. ostoyae* there were 3 Ftr1/Fip1/EfeU iron permease encoding genes (IPR004923) of which two were upregulated in the mycelium. Of these three iron permeases, two were accompanied by a multicopper oxidase gene (one had an upstream and the other had a downstream MCO to the Ftr1/FtrA iron permease respectively). In *A. cepistipes*, there were two iron permease encoding genes, of which one was upregulated in mycelium vs non-invasive mycelium. The genes downstream to these two iron permeases were multicopper oxidases, and both were upregulated in mycelium vs. non-invasive mycelium. Neither iron permeases nor the Fet3-multicopper oxidases were differentially expressed in rhizomorphs vs non-invasive rhizomorphs. In proteomics, we found 1 Ftr1/FtrA iron permease gene with increased abundance in mycelia of both *A. ostoyae* and *A. cepistipes*. There was one downstream multicopper oxidase gene with increased abundance in the mycelium only of *A. cepistipes* and not in *A. ostoyae*. The iron permease system along with iron reductases and MCOs is specifically seen upregulated in brown-rot wood decay [56,67,74,76,77,78].

### 3.9. Diverse Cytochrome P450s Are Differentially Expressed in Invasive Tissues

Cytochrome P450s have diverse functions across the fungal kingdom, including secondary metabolite production, detoxification, aromatic compound degradation, among others [79]. Based on InterPro domains (IPR001128) we found a total of 264 and 307 cytochrome P450 encoding genes in *A. ostoyae* and *A. cepistipes*, respectively. Of these 35 were upregulated and 57 downregulated in invasive mycelia along with three upregulated one downregulated in rhizomorphs of *A. ostoyae* relative to the non-invasive counterparts of these tissues. In proteomics data for *A. ostoyae*, we found seven increased and seven decreased cytP450s in M*vs*NIM, along with three increased and no proteins with decreased abundance in RvsNIR. In *A. cepistipes* we found 58 (proteomics: 13) up- and 84 (6) downregulated cytP450s in M*vs*NIM along with 27 (11) up- and 22 (7) downregulated cytP450s in RvsNIR. Most of the cytochrome P450s with increased abundance in invasive mycelia compared to non-invasive were classified as E-class group 1.

Interestingly, among the cytochrome P450 proteins that had decreased abundance in invasive mycelia were co-orthologs of the Psi-producing oxygenase A (*PpoA*) from *Aspergillus nidulans*, which is associated with secondary metabolite biosynthesis (sterigmatocystin), oxylipin biosynthesis, and coordination of a/sexual sporulation [80]. Ppo proteins are also implicated in virulence, with *A. fumigatus ppo* mutants displaying hypervirulence, possibly due to the activation of the immune response in mammals [81]. This is observed plant pathogens also, with *Fusarium verticillioides ppo* deletion strain showing higher virulence in assays with maize cobs as well as elevated fumonisin production and lower induction of plant defense-related genes [82]. Therefore, the downregulation of *PpoA* homologs in invasive mycelia of *A. ostoyae* and *A. cepistipes* may result in lower induction of plant defenses, alteration of secondary metabolism, and enhanced virulence.

### 3.10. Rhizomorphs Show an Upregulation of Transporters as Compared to Mycelium

*Armillaria* rhizomorphs are putatively involved in the translocation of nutrients [83]. Rhizomorphs are generally believed to serve as migratory organs for exploration of substrates across various distances, however, the studies also indicate rhizomorphs produced by saprotrophic Basidiomycetes are also effective in absorbing inorganic nutrients and water from the soil [84,85,86,87,88,89]. Experiments with *Armillaria mellea* [88,89] and *Serpula lacrymans* [90,91,92] demonstrated the translocation of various nutrients, water, and carbon within the rhizomorphs.

To investigate transporter expression in our wood-decay system, we classified putative transporters in *A. ostoyae* and *A. cepistipes* based on conserved domains. We identified 612 and 602 transporters in *A. ostoyae* and *A. cepistipes*, which belonged to 100 InterPro annotations with major facilitator superfamily domain, ABC-transporter like, sugar transporters, amino acid/polyamine transporters, and P-type ATPases being most abundant. Appendix A lists the identified transporters and their expression and abundances in the transcriptomic and proteomic data. Transcriptomics provided dynamics for a much larger number of transporters than proteomics, possibly due to the difficulty of extracting membrane proteins for LC/MS analyses. In the RNA-Seq data, we found 45 and 84 upregulated along with 96 and 105 downregulated transporters in mycelium vs non-invasive mycelium in *A. ostoyae* and *A. cepistipes,* respectively (Figure 6). The majority of transporters upregulated in the mycelium vs. non-invasive mycelium belonged to the major facilitator superfamily (MFS) and sugar transporter family. Considerably lower numbers of upregulated transporters were found in rhizomorphs: 2 and 20 upregulated in *A. ostoyae* and *A. cepistipes,* respectively, from the same families.

A striking difference in the expression of transporters was found between rhizomorphs vs mycelium (Figure 6), with 120 and 100 upregulated and much fewer downregulated transporters in *A. ostoyae* and *A. cepistipes,* respectively. Compared to the total number of these transporters, the most upregulated transporters in rhizomorphs vs. mycelium were the ones possibly involved in sugar transport, such as major facilitator sugar transport-like (IPR005828), sugar/inositol transporter (IPR003663) and sugar transporters (IPR005829). Several aquaporin-like proteins that were reported to be involved in mushroom development [93] and ectomycorrhizal functioning [94], were also upregulated in rhizomorph vs. mycelium.

We also compared the upregulated transporters in rhizomorphs vs. mycelium to fruiting body development regulated genes of *A. ostoyae* from Sipos et al. [17], to identify transporters that are specifically upregulated in rhizomorphs but not in fruiting bodies. We reasoned that such genes might be involved in rhizomorph-specific functions rather than shared multicellularity-related functions between rhizomorphs and fruiting bodies. We found 47 such genes (Appendix A), 39% of rhizomorph-upregulated transporters, including several MFS domains and sugar transporters. Collectively, our transporter data suggests rhizomorphs not being involved in active wood-decay rather they might be involved in the transfer of the decomposition intermediates between different parts of the colony.

## 4. Discussion

In this study, we compared wood-decay patterns by two species of *Armillaria* using transcriptomic and proteomic data on autoclaved spruce roots. Spruce roots were primarily colonized by fungal hyphae, which had a higher number of differentially expressed genes and differentially abundant proteins than did the rhizomorphs of both species. This suggests that although rhizomorphs can efficiently forage for nutrients, individual hyphae pierce and colonize woody tissues. Of the two species, the saprotroph, *A. cepistipes,* showed a much stronger transcription- and protein-level response to wood than did the pathogenic *A. ostoyae*. This pattern was evident both in terms of the number of differentially expressed genes/proteins and in gene expression dynamics (e.g., fold change) displayed by the two species. We observed a higher number of upregulated PCWDEs in *A. cepistipes* than in *A. ostoyae* with the latter species showing more down- than upregulation in PCWDEs. We speculate that these observations might reflect a general difference in wood-decay strategies of saprotrophic vs. pathogenic species. Because saprotrophs colonize dead wood, they likely face more intense competition with other microbes than do necrotrophic pathogens, which, after killing the host, are the very first colonizers, and thus might face less competition. This might select for more rapid colonizing and substrate-degrading wood-decay strategy in saprotrophs, which, in an assay like ours, may manifest as a stronger induction of specific CAZymes and other proteins. In comparison, pathogens, which can also feed while the host is alive, may not be under a strong pressure to express a large suite of wood-decay enzymes. Similar observations were made in a study comparing gene expression during saprotrophic and parasitic phases in *Heterobasidion irregulare* [29]. It should be noted that these observations might also be influenced by the substrate (though spruce is a natural substrate for both species), the individual properties of the strains, and other factors, so more evidence, and experimental testing are needed to confirm this hypothesis. Another thing to be noted is that this study was aimed at comparing the changes during wood decay in a pathogenic and saprotrophic *Armillaria* species, further time-series wood decay analysis could prove to be helpful in understanding the wood-decay dynamics in *Armillaria* in detail.

Mycelia of *A. ostoyae* and *A. cepistipes* responded similarly to wood, with 713 orthologous genes showing differential expression in both species. These include many plant cell wall degrading enzyme genes, hydrophobins, CBM67s, cytochrome p450s, transcription factors, and iron acquisition-related genes, among others. Pectinolytic PCWDE genes were dominant among CAZymes, followed by expansins, and cellulose- and hemicellulose degrading enzyme genes, whereas lignin-degrading PCWDE-s were mostly downregulated in our assay. The proportionately high number of pectin-related PCWDE-s mirrors comparative genomic observations that revealed enrichment of pectinolytic genes (in particular CBM67s) and expansins, but depletion of LDE encoding genes in *Armillaria* genomes [17]. Of particular interest are CBM67-s, which comprised 4 of the top 20 most induced co-orthologous genes in both species. There was a similar percentage of upregulated pectinolytic genes in the mycelium of *A. cepistipes* (ca. 38%) as in other studies examining early-phase decay (48% in *Pycnoporus coccineus* [95] and 45% in *Postia placenta* [67]), whereas the percentage was lower in *A. ostoyae* (17%). LDE encoding genes (AA1, AA2, and AA5), on the other hand, were underrepresented among DEGs/DAPs, consistent with their general underrepresentation in *Armillaria* genomes. This aligns well with previous reports of the limited lignin-degrading capacity of *Armillaria* spp. [22,23,96].

Several aspects of the gene expression patterns in our assays are unusual for white-rot fungi. These include the lack of an early burst of LDE encoding gene expression as is typical for white-rot fungi and the high expression of some genes (e.g., iron uptake systems, oxoglutarate/iron-dependent dioxygenases) that have been reported from brown rot fungi [74,76,77,78]. Previous studies questioned the typical white-rot nature of *Armillaria* [20,22,23,96]. *Armillaria mellea* was reported to attack celluloses but not lignin in the early stages of decay [23,96]. We previously showed that, based on LDEs, *Armillaria* spp. are intermediate between brown-rot and white-rot fungi, along with other species that have been recalcitrant to the WR/BR classification [97], as well as ectomycorrhizal fungi with reduced LDE encoding gene repertoires. Of these, *Cylindrobasidium torrendii* is a close relative of *Armillaria* in the Physalacriaceae. Floudas et al. [20] showed that the reduced LDE encoding gene repertoire of *C. torrendii* is a result of gene loss compared to its white-rot ancestors and that the decay of dead wood caused by this species resembles soft rot. While soft rot is characteristic to Ascomycota, especially in the class Sordariomycetes, several Basidiomycetes have been associated with Type II (e.g., *C. torrendii* [20]) or Type I (e.g., *Armillaria* spp. [22]) soft rot. These species are characterized by a complete set of PCWDEs for degrading cellulose and hemicellulose, but a depletion of genes encoding LDEs. More generally, a selective deployment of lignin-degrading activity has been reported in a number of wood-decaying fungal species of Basidiomycota. *Mucidula mucida* (of the order Agaricales, as *Oudemansiella mucida)* and *Meripilus giganteus* (of the order Polyporales) caused either soft or white rot, depending on decay stage and substrate (host species, and type of wood and plant cell composition) [98,99]. Recently, *Pleurotus ostreatus,* a species typically reported to cause white rot, was reported to produce soft-rot like symptoms [100].

Wood decay resembling soft rot has been reported also for several early-diverging Agaricomycetes (e.g., in non-mycorrhizal Cantharellales), which predate the origin of lignin-degrading class II peroxidases [7,8]. We hypothesized that these early diverging lineages of Basidiomycota class Agaricomycetes, together with a few later diverged species that have lost their lignin-attacking genes and ability (e.g., *Jaapia* and *Schizophyllum*), reverted to a plesiomorphic soft-rot like decay mechanism, which is primarily dominated by cellulolytic and pectinolytic functions [8]. It appears that this decay profile is widespread among Agaricomycetes, and we suggest that such species represent secondarily soft-rot fungi that have partially lost their ability to enzymatically attack and degrade lignin due to depletion and/or disability to produce catalytically efficient class-II peroxidase encoding. We think that the combination of (i) a widely conserved plesiomorphic soft rot-like wood decay strategy and (ii) the ability to degrade lignin in white rot, enables Agaricomycetes to toggle between soft- and white rot either by gene loss or by gene expression regulation. Thus, it is possible that temporal or substrate-dependent regulation of the activation of LDE encoding genes can separate soft- and white- rot behaviors of some species, adding further complexity to the range of decay modes of Basidiomycota.

## Figures and Tables

**Figure 1 microorganisms-09-00149-f001:**
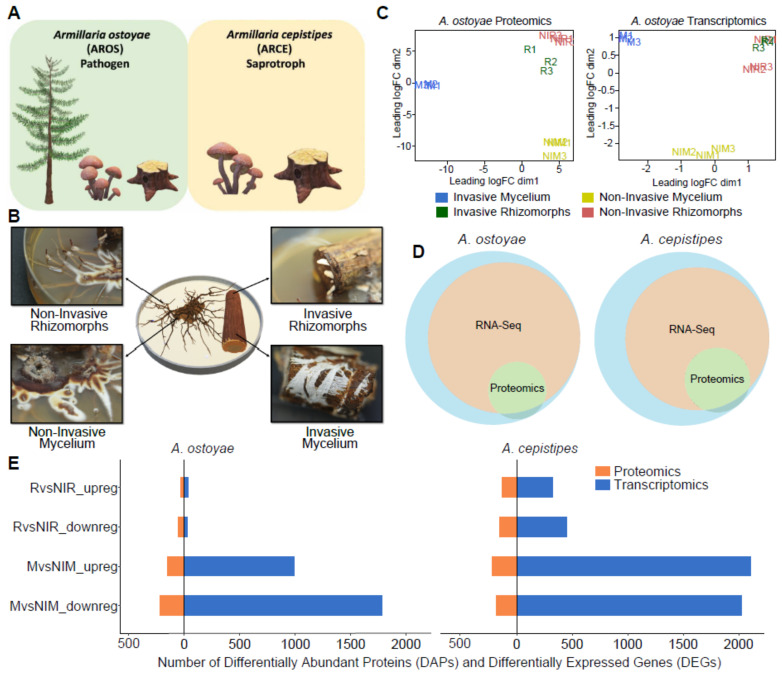
Overview of the experimental approach for root decay studies. (**A**) Representation of *Armillaria ostoyae* (pathogenic) and *A. cepistipes* (saprotroph) used in this study, (**B**) The four tissue types sampled for transcriptomics and proteomics analysis, viz., invasive mycelium (growing beneath the outer layer of root), invasive rhizomorphs (emerging out of the roots), non-invasive mycelium and non-invasive rhizomorphs (growing in absence of root). *A. ostoyae* is shown as an example, (**C**) Multidimensional scaling of three biological replicates from each of the tissue types in *A. ostoyae* for proteomics (left) and transcriptomics (right), (**D**) Proportion of transcripts and proteins detected in the two -omics analysis. The blue circle represents the whole proteome, orange depicts the transcripts detected in the RNASeq, and green represents the proteins detected in the proteomics analyses. (**E**) The number of differentially expressed genes (blue) and differentially abundant proteins (orange) detected in the two species.

**Figure 2 microorganisms-09-00149-f002:**
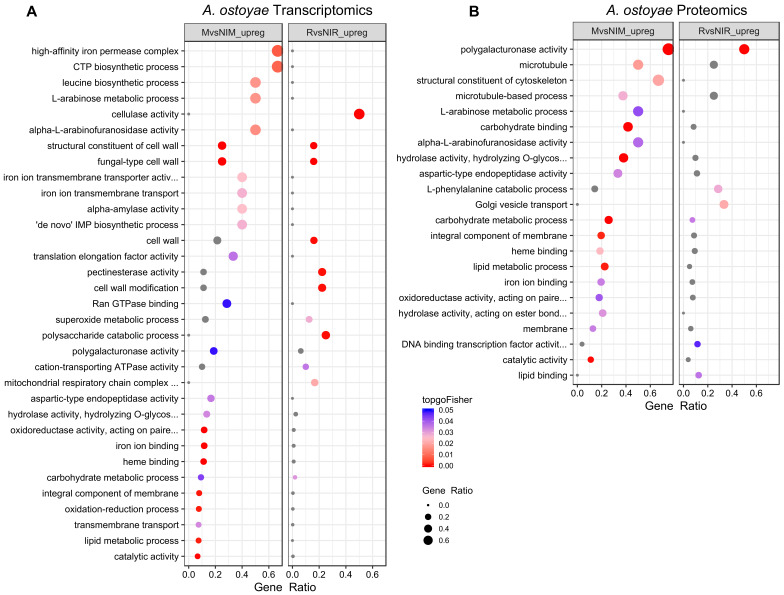
Enriched GO terms in M*vs*NIM and R*vs*NIR of *A. ostoyae* for transcriptomics (**A**) and proteomics (**B**). The ratio of number of a particular GO term in a specific comparison (mycelium vs. non-invasive mycelium or in rhizomorphs vs non-invasive rhizomorphs) to the total number of that GO term for a species was used to plot gene ratios for enriched GO terms (*p* < 0.05, Fisher’s exact test). The size of the dot is directly proportional to gene ratio, and the color of the dots corresponds to *p*-values. Grey dots represent GO terms, enriched in only one of the comparisons i.e. either mycelium vs. non-invasive mycelium or rhizomorphs vs. non-invasive rhizomorphs. Enriched GO terms in M*vs*NIM and R*vs*NIR of *A. cepistipes* can be found in Appendix A.

**Figure 3 microorganisms-09-00149-f003:**
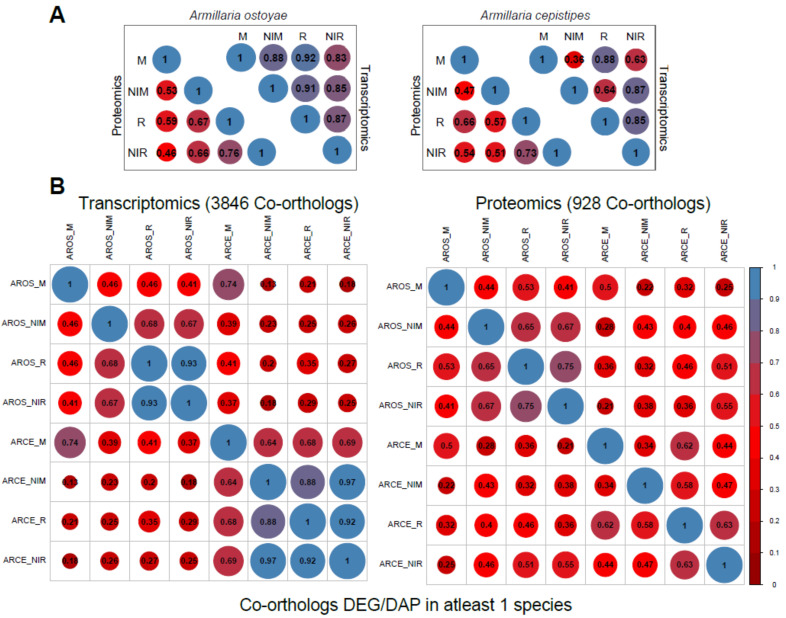
Global transcriptome/proteome similarity between *A. ostoyae* and *A. cepistipes*. (**A**) Correlation between the 4 tissue types for proteomics and transcriptomics data in *A. ostoyae* (left) and *A. cepistipes* (right). (**B**) Correlation between co-orthologs which were significantly differentially expressed/abundant in at least one of the species (for all co-orthologs, see Appendix A), showing correlation between samples across the two species. Blue represents higher correlation and red represents lower. Size of the circle is directly proportional to higher correlation. Pairwise mean Pearson correlation coefficients are given as numbers in the circles.

**Figure 4 microorganisms-09-00149-f004:**
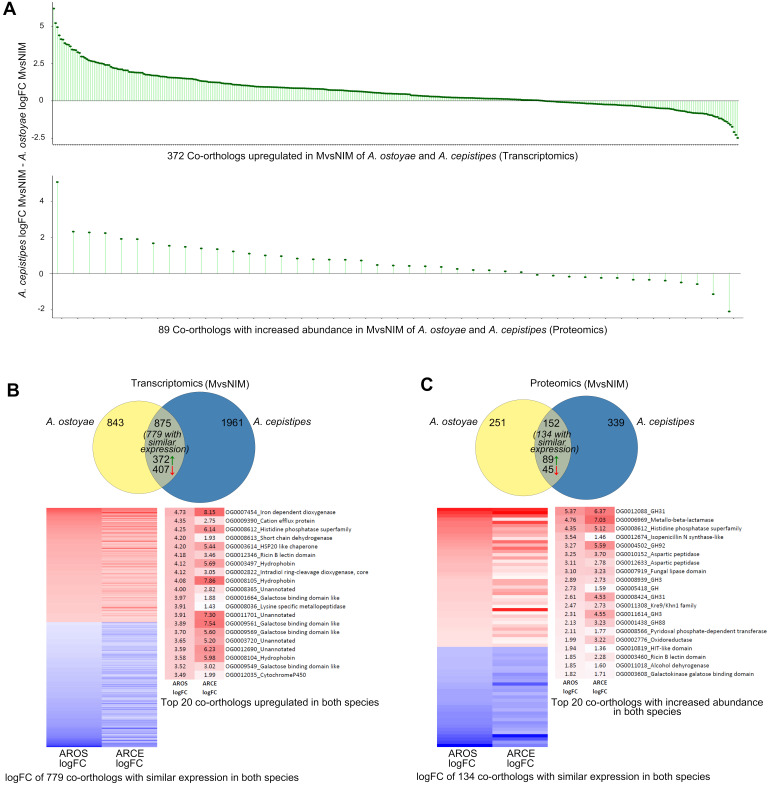
Response towards spruce roots by the mycelia (MvsNIM) of *A. ostoyae* and *A. cepistipes*. (**A**) logFC differences of co-orthologs DEG (top) and DAP (bottom) in both species. For proteomics, only genes for which a fold change could be calculated are shown (43 out of 89 orthologs). (**B**) Venn diagram showing species-specific and common DEGs in *A. ostoyae* and *A. cepistipes*. Green and red arrows correspond to upregulated and downregulated genes, respectively. Heatmap below shows 779 co-orthologs with similar expression patterns (upregulated/downregulated in both species) out of the 875 common DEGs. Towards right are the top 20 upregulated proteins (red) in the two species, sorted by decreasing logFC in *A. ostoyae*. (**C**) Venn diagram showing species-specific and common DAPs in *A. ostoyae* and *A. cepistipes*. Green and red arrows correspond to proteins with increased and decreased abundance, respectively. Heatmap below shows 134 co-orthologs with similar abundance (increased/decreased in both species) out of the 152 common DAPs. Towards right are the top 20 proteins with increased abundance in the two species, sorted by decreasing logFC in *A. ostoyae*.

**Figure 5 microorganisms-09-00149-f005:**
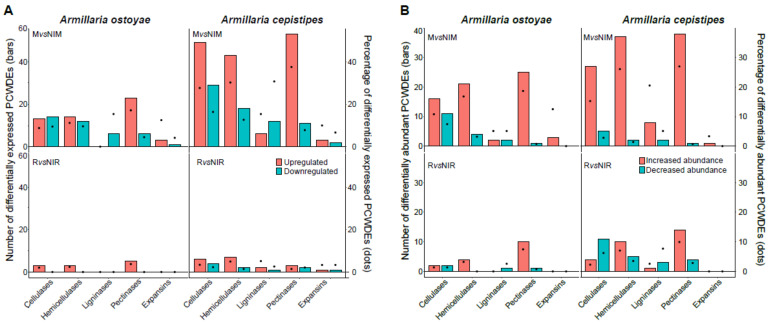
Differentially expressed/abundant plant cell wall degrading enzymes in the two species. (**A**) Barplot showing number, and dots showing the percentage of differentially expressed PCWDEs in the two species in MvsNIM (top) and RvsNIR (bottom) in transcriptomics data. (**B**) Barplot showing the number, and dots showing the percentage of differentially abundant PCWDEs in the two species in MvsNIM (top) and RvsNIR (bottom) in proteomics data. CAZYmes classified on the basis of their substrate in the plant cell wall (Appendix A) showing the number of genes upregulated/increased abundance (orange) and downregulated/decrease (blue-green) in the two species.

**Figure 6 microorganisms-09-00149-f006:**
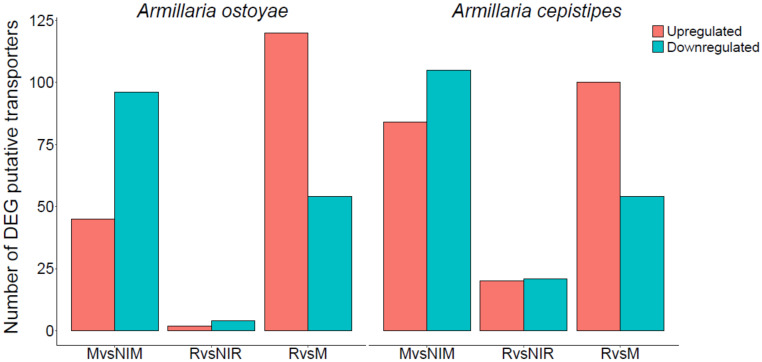
Number of differentially expressed putative transporters in the two species. The number of upregulated (orange) and downregulated (blue-green) genes are shown for M*vs*NIM, R*vs*NIR, and R*vs*M comparisons in the two species.

## Data Availability

The RNA-Seq data was deposited in the NCBI’s Gene Expression Omnibus (GEO) Archive at 560 www.ncbi.nlm.nih.gov/geo (accession no. GSE149732). Other data used in this study are available on request from the corresponding author.

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
