# Peer review of "Hallmarks of Basidiomycete Soft- and White-Rot in Wood-Decay -Omics Data of Two Armillaria Species"

_microorganisms, 2021, doi:10.3390/microorganisms9010149_

Round 1

Reviewer 1 Report

The paper is a great study and description of transcriptome and proteome machineries of Armillaria, however the authors based their conclusions on short laboratory experiments. In this paper the sterilized roots of Norway spruce were treated with one week old cultures of A. ostoyae and A. cepistipes, which differ in their infectiousness. The cultures were co-cultured for 2-3 weeks only (the authors have not given the precise time) and then the observations and subsequent conclusions were done using the collected material. The authors have noted the abundant growth of the mycelium, but no rhizomorphs. This is not a surprise at all, as A. ostoyae extremely rarely produces rhizomorphs under laboratory conditions. Also the fungus in natural condition produces rhizomorphs only when the bark and the wood are becoming separate. Indeed in natural conditions colonization between the tree roots is done by the mycelium. The Authors seem to have a very good knowledge of molecular techniques, but the study is based on grounds which may lead to misinterpretation of the results. I suggest to undertake the decision of the editors after contacting the specialist on Armillaria.

Author Response

We thank Reviewer 1 for their valuable suggestions towards the improvement of the manuscript. Please find below our reasoning for the comments:

  • The cultures were co-cultured for 2-3 weeks only (the authors have not given the precise time) and then the observations and subsequent conclusions were done using the collected material.

Ans: We added a more precise definition of the time point we used (L.510): We incubated the Armillaria cultures until the roots were completely colonized and we observed rhizomorphs growing out of them. Sampling was made at the time point of rhizomorphs emerging out of the root pieces.

  • The authors have noted the abundant growth of the mycelium, but no rhizomorphs. This is not a surprise at all, as A. ostoyae extremely rarely produces rhizomorphs under laboratory conditions. Also the fungus in natural condition produces rhizomorphs only when the bark and the wood are becoming separate. Indeed in natural conditions colonization between the tree roots is done by the mycelium.

Ans: We do not agree with the given comment. The C18 strain of Armillaria ostoyae (Sipos et al 2017) has been shown to produce abundant rhizomorphs under laboratory conditions in undefined media (such as Malt extract, potato dextrose, RTSO media). In our experiments as well, on Malt extract agar, A. ostoyae produced rhizomorphs, both on control plates and plates which were introduced with spruce roots. However, rhizomorphs were not found under the bark layer when they were dissected, but only found growing aerially out of the colonized bark. 

  • The Authors seem to have a very good knowledge of molecular techniques, but the study is based on grounds which may lead to misinterpretation of the results.

Ans: We are not sure we understand which section(s) the Reviewer has in mind here, we would welcome more specific suggestions on where the Reviewer feels we misinterpreted our observations. Our team includes specialists of Armillaria (G. Sipos), with several years of experience with these organisms.

Reviewer 2 Report

The manuscript of Sahu et al. presents a novel study to understand wood-decay strategies of two Armillaria species, one saprotroph, the other pathogenic, using a combined transcriptomic and proteomic approach. An in vitro system with Norway spruce roots was used and both invasive and non-invasive mycelia as well as rhizomorphs were investigated. 
This is very interesting, the analyses overall well performed and described in this draft.

Unfortunately there is no chemical description of the decomposed material. For example the initial cell wall composition and at the end of the experiment, to link the gene expression and proteomic data to the actual degradation. Eventually adding also more microscopy images to compare the different impact of the two fungi?
There is also only one time-point analyzed, which unfortunately makes statements about eventual successive induction of enzymes impossible.

Further points to address:
L122 Could the poor mapping be due to remaining ribosomal reads or other contaminations since a ribo-zero approach was used? Please give more information about the origin of the non-mapping (majority) of sequences!
L154 Can you exclude that just higher variations in biological replicates influence the number of significantly differentially expressed genes?
L319 As already stated, it would be important to link cell wall composition to the data.

L516 Why the authors opted for a Ribo-Zero approach? Both organisms are eukaryotes. Would a mRNA based sequencing approach be not more appropriate?

Author Response

We thank Reviewer 2 for their valuable suggestions towards the improvement of the manuscript. Please find below our reasoning for the comments:

  • Unfortunately there is no chemical description of the decomposed material. For example the initial cell wall composition and at the end of the experiment, to link the gene expression and proteomic data to the actual degradation. Eventually adding also more microscopy images to compare the different impact of the two fungi?

Ans: We have added information on chemical composition of spruce (L.303) and recent results on chemical and microscopic analysis during wood decay (L.485). We agree with the reviewer, that analyzing the chemical composition of the roots would add additional perspective to our understanding of the process. We are unfortunately not in a position to perform the kinds of experiments that would be needed, partially due to the lockdown and partially to the lack of appropriate instrumentation. 

  • There is also only one time-point analyzed, which unfortunately makes statements about eventual successive induction of enzymes impossible.

Ans: We agree that analyzing multiple time points would have been beneficial, but the emphasis of the study is on the comparative analysis of A. ostoyae and A. cepistipes. Therefore, in our study design we opted for single time points across two species. (This has been mentioned in the ms at L.87)

Further points to address:

  • L122 Could the poor mapping be due to remaining ribosomal reads or other contaminations since a ribo-zero approach was used? Please give more information about the origin of the non-mapping (majority) of sequences!

Ans. We found that the poor mapping could be a result of genomic DNA contamination. This is based on mapping the reads to both the genome and to predicted CDS. We found that mapping to the genome yielded a mapping of ~65-70%, which is typical for RNA-Seq studies (including ~50% in intergenic regions), which led us to conclude that genomic DNA contamination caused the low mapping percentages. Nevertheless, as we mention in the manuscript, we are confident that this did not undermine our conclusions on differentially expressed genes and on the comparison of A. cepistipes and A. ostoyae. Initial quality checks of our data (PHRED scores, GC tailing) indicated that the reads are of high quality, similar to those obtained for A. ostoyae. 

  • L154 Can you exclude that just higher variations in biological replicates influence the number of significantly differentially expressed genes?

Ans. Yes, we are confident that higher variance among replicates did not cause differences in DEG numbers. For both species, biological replicates grouped tightly together, as shown by hierarchical clustering analysis and the MDS plots on Supplementary Figure 1A

  • L319 As already stated, it would be important to link cell wall composition to the data.

Ans. please see our answer above.

  • L516 Why the authors opted for a Ribo-Zero approach? Both organisms are eukaryotes. Would a mRNA based sequencing approach be not more appropriate?

Ans. We agree that an mRNA-based library would have been equally appropriate. We initially opted for the Ribo-Zero approach with the intention to retain non-polyA transcripts and to reconstruct potential long noncoding RNA. However, the sequencing provider omitted strand-specificity from the library preparation, which left us with data that is equivalent to a conventional mRNA-based library. We did not detect ribosomal RNA contamination in our data.

Round 2

Reviewer 1 Report

The corrections in Materials and methods have made the text clear. If the mycelium was covered with spruce roots, and rhizomorphs were created from the mycelium which has overgrown the wood, all is perfectly OK. Armillaria ostoyae very rarely forms rhizomorphs on an artificial solid medium (Malt, PDA), but often on a liquid medium, and when wood with bark is added, the fungus is able to produce such rhizomorphs. I was afraid the Authors  want to prove that rhizomorphs have nothing in common with the infection process, this would be a nonsense. From the work I understand a completely different idea that rhizomorphs do not take part in the decomposition of wood. This is likely, so the conclusions made by the Authors have good grounds. I have no objections to the genetic/molecular part of the work. As such, I think the manuscript is really good and I have no objections any longer. The previous ones originated from an understatement and subsequent misunderstanding. I suggest to accept the manuscript in its current form, after the necessary editorial changes/corrections, already listed by the Editor.